# Model Supply Chain Poisoning: Backdooring Pre-trained Models via Embedding Indistinguishability

Submission Id: 1525

## Abstract

Pre-trained models (PTMs) are widely adopted across various downstream tasks in the machine learning supply chain. Adopting untrustworthy PTMs introduces significant security risks, where adversaries can poison the model supply chain by embedding hidden malicious behaviors (backdoors) into PTMs. However, existing backdoor attacks to PTMs can only achieve partially task-agnostic and the embedded backdoors are easily erased during the fine-tuning process. This makes it challenging for the backdoors to persist and propagate through the supply chain. In this paper, we propose a novel and severer backdoor attack, TransTroj, which enables the backdoors embedded in PTMs to efficiently transfer in the model supply chain. In particular, we first formalize this attack as an indistinguishability problem between poisoned and clean samples in the embedding space. We decompose embedding indistinguishability into pre- and post-indistinguishability, representing the similarity of the poisoned and reference embeddings before and after the attack. Then, we propose a two-stage optimization that separately optimizes triggers and victim PTMs to achieve embedding indistinguishability. We evaluate TransTroj on four PTMs and six downstream tasks. Experimental results show that our method significantly outperforms SOTA task-agnostic backdoor attacks – achieving nearly 100% attack success rate on most downstream tasks – and demonstrates robustness under various system settings. Our findings underscore the urgent need to secure the model supply chain against such transferable backdoor attacks. The code is available at https://anonymous.4open.science/r/TransTroj.

## CCS Concepts

• **Computing methodologies** → **Computer vision**.

## Keywords

Backdoor attack, Pre-trained model, Model supply chain

## 1 Introduction

Pre-trained models (PTMs) have revolutionized the field of machine learning, serving as foundational elements that can be fine-tuned for a wide range of downstream tasks [4–6, 9, 15, 16, 18, 27]. By leveraging extensive datasets and substantial computational resources during the pre-training phase, PTMs enable practitioners to achieve remarkable performance without the need to train models from scratch. Consequently, developers frequently source PTMs from public repositories such as Hugging Face and GitHub to expedite their development cycles.

Unfortunately, this reliance on externally sourced PTMs introduces significant security vulnerabilities into the machine learning supply chain. Specifically, incorporating untrustworthy PTMs can expose applications to backdoor attacks, where adversaries embed hidden malicious behaviors within the models [1, 3, 7, 12, 14, 23, 24, 35–37]. These backdoors remain dormant under normal conditions but can be activated by specific triggers, causing the model to perform unintended actions that serve the attacker's interests. This form of **model supply chain poisoning** poses a critical threat, as compromised PTMs can proliferate across various systems and domains, leading to widespread security breaches.

Existing backdoor attacks targeting pre-trained models are usually task-specific, often implemented by poisoning the training data of specific downstream tasks. This poisoning involves inserting triggers into the data and modifying labels, which requires prior knowledge of downstream tasks—including specific datasets, labels, or training configurations [20, 21, 38, 40]. Others depend on particular pre-training strategies, such as contrastive learning, which limits their applicability [2, 3, 10, 28, 30, 39].

Task-agnostic backdoor attacks have emerged to address these constraints [17, 30, 33, 41]. Due to the absence of downstream labels, the key to achieve task-agnostic backdoor attacks lies in aligning a backdoor trigger with the pre-defined reference embedding. Such embedding can hit a target label of various downstream tasks after fine-tuning. The reference embedding is usually predicted from the target class images (shadow images) [17, 33] or created manually using empirical methods [30, 41]. The manually created reference embeddings are also known as artificially pre-defined output representations (PORs).

However, these task-agnostic backdoor attacks still face two critical challenges: **(1) Durability**: The embedded backdoors are susceptible to being erased during the fine-tuning process due to catastrophic forgetting [25]. The association between triggers and target behaviors is fragile, especially when triggers are unlikely to appear in downstream datasets. **(2) Partial Task-Agnosticism**: These attacks can only partially generalize across tasks. For instance, artificially pre-defined output representations (PORs) used in some methods may not correspond to the attacker's intended target class across different tasks, leading to inconsistent backdoor activation.

In this paper, we introduce **TransTroj**, a novel backdoor attack that overcomes these limitations by embedding backdoors into PTMs in a manner that efficiently transfers through the model supply chain. Our key innovation lies in formalizing the attack as an *embedding indistinguishability* problem between poisoned and clean samples within the embedding space of the model. By ensuring that poisoned samples are indistinguishable from clean samples of a target class in the embedding space, we create a backdoor that is both durable and task-agnostic. To achieve this, we decompose embedding indistinguishability into two components: **(1) Pre-Indistinguishability**: The similarity between the embeddings of poisoned samples and reference embeddings before the attack. **(2) Post-Indistinguishability**: The similarity between these embeddings after the attack, ensuring that the backdoor effect persists

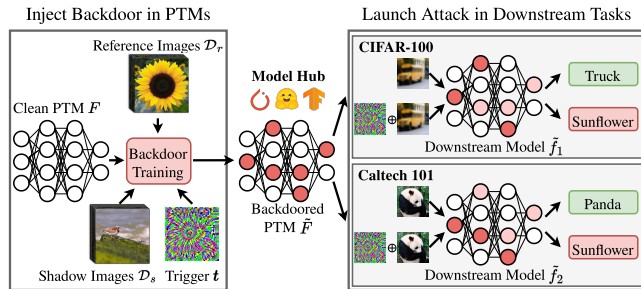

**Figure 1: Illustration of transferable backdoor attacks. The adversary injects backdoor into a clean PTM and launches attack when the backdoored PTM is leveraged to fine-tune downstream tasks. Note that the task-agnostic backdoor can be activated in various downstream tasks.**

post-fine-tuning. We then design a two-stage optimization framework to meet these objectives: **(1) Trigger Optimization**: We generate a universal trigger by aggregating embeddings from publicly available samples of the target class. This trigger is optimized to enhance pre-indistinguishability, making poisoned samples mimic the reference embeddings. **(2) Victim Model Optimization**: We fine-tune the victim PTM on a carefully crafted poisoned dataset to reinforce post-indistinguishability while preserving the model's performance on clean data.

We conduct extensive evaluations of TransTroj on four widely used PTMs – ResNet, VGG, ViT, and CLIP – and six downstream tasks, including CIFAR-10, CIFAR-100, GTSRB, Caltech 101, Caltech 256, and Oxford-IIIT Pet. Our experimental results demonstrate that TransTroj significantly outperforms SOTA task-agnostic backdoor attacks, achieving nearly 100% attack success rates on most downstream tasks with minimal impact on model accuracy. Moreover, TransTroj exhibits robustness under various system settings and remains effective even when subjected to model reconstruction-based defenses.

Our key contributions are summarized as follows:

- We propose **TransTroj**, a novel backdoor attack that is functionality-preserving, durable, and truly task-agnostic, effectively transferring through the model supply chain.
- We introduce the concept of *embedding indistinguishability* and decompose it into pre- and post-indistinguishability to systematically craft durable and transferable backdoors.
- We design a two-stage optimization framework that separately optimizes the trigger and the victim PTM to achieve embedding indistinguishability without sacrificing model performance on clean data.
- We provide comprehensive experimental evidence demonstrating the effectiveness and robustness of TransTroj across multiple PTMs and downstream tasks, achieving nearly 100% attack success rates on most downstream tasks.

## 2 Related Work and Comparisons

Existing backdoor attacks against PTMs can mainly be categorized into two types: task-specific [2, 3, 10, 20, 21, 28, 31, 31, 38–40] and task-agnostic [17, 33, 41], as summarized in Tab. 1.

**Task-specific backdoor attacks.** Existing task-specific backdoor attacks [20, 21, 38, 40] against PTMs usually require prior knowledge of the downstream tasks. For example, Kurita *et al.* [20] proposed RIPPLe, which directly poisons the fine-tuned downstream model and then acquires the PTM part as a task-specific backdoored PTM. Yao *et al.* [38] proposed a latent backdoor attack (LBA) to inject backdoors into a teacher classifier built on PTMs. To successfully attack the specific downstream task, LBA also needs a labeled dataset that is similar to the target downstream dataset. When the backdoored teacher classifier is used to fine-tune a student classifier for the specific target downstream task, the student classifier inherits the backdoor behavior. Other backdoor attacks are designed for specific pre-training paradigms like contrastive learning [2, 28, 39], masked image modeling [31], and masked language modeling [3, 10, 31].

**Task-agnostic backdoor attacks.** Compared to task-specific attacks, task-agnostic attacks [17, 33, 41] are more severe threats as they can compromise various downstream tasks built on the PTMs. BadEncoder[17] exemplified a backdoor attack on self-supervised learning, which compromises the downstream classifier by inserting backdoors into an image encoder. It achieves a high attack success rate in transfer learning scenarios without fine-tuning PTMs, such as linear probing and zero-shot classification. NeuBA [41] trained PTMs to build strong links between triggers and manually pre-defined output representations (PORs). After fine-tuning, the POR can hit a certain label of the downstream task. In binary classification tasks, the PORs of NueBA can cover positive and negative classes and accomplish targeted attacks. Unfortunately, existing attacks are unable to maintain a high attack success rate after fine-tuning or be fully task-agnostic. To the best of our knowledge, TransTroj is the first method that simultaneously satisfies functionality-preserving, durable, and task-agnostic.

## 3 Problem Statement

### 3.1 System Model

We consider the pre-training-then-fine-tuning paradigm in this study. Fig. 1 illustrates the system model, where the Model Publisher (**MP**) first trains or fine-tunes a PTM $F$ using his/her own training data and releases the weights $\theta^F$ to model hubs like Hugging Face. Second, the Downstream User (**DU**) downloads the publicly available PTM and adapts it to fit his/her downstream tasks. For a classification downstream task, the corresponding model $f$ is typically built by appending a linear classification head to $F$. To ensure the downstream model operates as anticipated, **DU** fine-tunes the entire model using the downstream dataset $\mathcal{D}_t$. The optimization objective can be formalized as:

$$\min_{\theta^F, \theta^h} \{\mathbb{E}_{(\boldsymbol{x}, y) \in \mathcal{D}_t} \mathcal{L}(f(\boldsymbol{x}), y)\}, \tag{1}$$

where $\theta^h$ refers to the classification head weights of $f$. The pair $(\boldsymbol{x}, y)$ signifies a sample from the downstream dataset $\mathcal{D}_t$. The loss function, symbolized as $\mathcal{L}$, is conventionally implemented with cross-entropy loss for classification tasks. Note that the PTM weights $\theta^F$ are adjusted to align the downstream data during the fine-tuning process.

**Table 1: Comparison of backdoor attack methods on pre-trained models.** *Prior knowledge*: the attack requires specific downstream task knowledge; *Task-agnostic*: backdoors can be activated across various downstream tasks; *Durable*: the attack maintains a high attack success rate even after fine-tuning.

| Method | Publication | Method Basis | Trigger Pattern | Prior Knowledge | Task-Agnostic | Durable |
|---|---|---|---|---|---|---|
| BadNets [14] | arXiv'17 | Data Poisoning-Based | Patch | Yes | No | No |
| CorruptEncoder [39] | CVPR'24 | Data Poisoning-Based | Patch | Yes | No | No |
| LBA [38] | CCS'19 | Weight Poisoning-Based | Patch | Yes | No | No |
| RIPPLe [20] | ACL'20 | Weight Poisoning-Based | Rare word | Yes | No | No |
| BadEncoder [17] | S&P'22 | Weight Poisoning-Based | Patch | No | Yes | No |
| NeuBA [41] | MIR'23 | Weight Poisoning-Based | Patch | No | Partially | No |
| TransTroj (Ours) | – | Weight Poisoning-Based | Optimized | No | Yes | Yes |

## 3.2 Threat Model

**Attack scenarios.** Fig. 1 shows the overall pipeline of weight poisoning backdoor attack against PTMs. Due to the increasing of computational overhead for pre-training a model from scratch, regular users tend to download PTMs from open-source repositories (*e.g.*, Hugging Face). However, this practice provides an opportunity for backdoor attacks, where adversaries can embed backdoors into models and upload them to open-source platforms. They may then employ tactics like URL hijacking to trick unsuspecting users into downloading these compromised models.

**Adversary's goal.** In our threat model, the model publisher **MP** is malicious, where he/her trains and releases an evil PTM, denoted as $\tilde{F}$, with a "transferable" backdoor. The goal is that any downstream model $\tilde{f}$ developed based on $\tilde{F}$ will inherit the backdoor. As shown in Fig. 1, the adversary initially prepares a clean PTM $F$. He/she selects a target classes $y_t$ and optimizes the corresponding trigger $t$. Then, he/she crafts a backdoored PTM $\tilde{F}$ to bind the trigger $t$ with the target class $y_t$. When a victim **DU** downloads the backdoored PTM $\tilde{F}$ and adapts it to his/her downstream tasks through fine-tuning, the adversary can activate the backdoor in the downstream model by querying with samples that contains the trigger $t$. A transferable backdoor should meet the following goals:

(1) *Functionality-preserving*. The malicious PTM $\tilde{F}$ should still preserve its original functionality. In particular, the downstream model $\tilde{f}$ built based on the backdoored PTM $\tilde{F}$ should be as accurate as the downstream model $f$ constructed based on the clean PTM $F$.

(2) *Durable*. Fine-tuning the PTM weights can lead the model to forget some previously learned knowledge, a phenomenon known as catastrophic forgetting [25]. A transferable backdoor must resist being forgotten during the fine-tuning process.

(3) *Task-agnostic*. A transferable backdoor should be effective for any downstream task instead of a specific one. In particular, when the target class $y_t$ is included in the downstream task, $\tilde{f}$ should predict any input $x$ with the trigger $t$ as $y_t$, *i.e.*, $y_t = \tilde{f}(x \oplus t)$, where $\oplus$ is the operator for injecting trigger $t$ into the input $x$.

**Adversary's capability.** We minimize the adversary's resources to make the attack more threatening and practical. (i) *A clean PTM and publicly available unlabeled images.* The adversary possesses full knowledge of the PTM, including the model structure and weights.

To carry out backdoor training, the adversary has access to a collection of images, designated as shadow dataset $\mathcal{D}_s$. Considering shadow images do not necessitate manual annotations, they can be effortlessly collected from the Internet. Meanwhile, the adversary needs a small set of reference images $\mathcal{D}_r$ for each target class, which can also be downloaded from the Internet. (ii) *None prior knowledge of downstream task*. The adversary has no prior knowledge of the downstream tasks. He/she lacks access to the downstream datasets and cannot manipulate the fine-tuning process.

## 4 Methodology

### 4.1 Observations and Pipeline

We aim to devise a transferable backdoor attack to fulfill all the adversarial goals. Our methodology is primarily derived from the following two observations.

**Observation I.** Existing attacks [17, 33, 41] typically use hand-crafted backdoor triggers, such as a patch located at the bottom-right corner of an input image. As trigger patterns are absent in the downstream data, these backdoors are prone to being forgotten during the fine-tuning process. Conceptually, if the trigger exhibits semantic similarity with the target class, then the samples of the target class in the downstream data could provide an avenue to sustain the backdoor.

**Observation II.** Several studies [10, 30, 41] bind triggers to PORs. However, as shown in Fig. 2, even when multiple PORs are embedded in the PTM concurrently, they do not cover the target class (*i.e.*, Dog) because of a lack of prior knowledge of downstream tasks. Fortunately, images of the target class (*i.e.*, reference images) can be easily downloaded from the Internet. The corresponding reference embeddings are good approximations of the embeddings of the downstream target class.

**Insight and pipeline.** Our key intuition is that a transferable backdoor attack should render poisoned inputs indistinguishable from clean inputs within the embedding space. For example, should the PTM generate an embedding for a poisoned sample identical to that of a dog image, the downstream model would misclassify the poisoned sample as a dog. Hence, our backdoor attack aims to ensure the poisoned samples' embeddings closely resemble those of the target class samples. Our attack pipeline is illustrated in Fig. 3. Given the inaccessibility to target class images in downstream tasks, the adversary procures reference images from the Internet to estimate reference embeddings. The adversary then strives to augment the similarity between the poisoned and the reference embeddings.

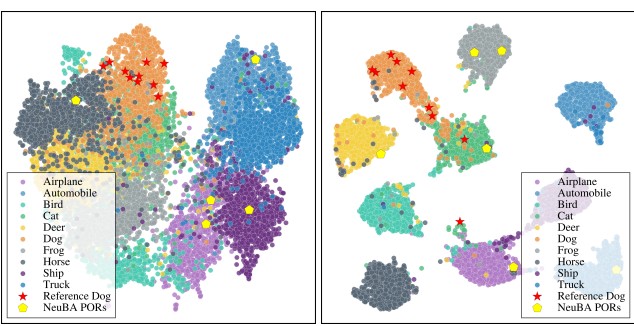

(a) Pre-trained Model      (b) Downstream Model

**Figure 2: Visualization of dimension-reduced embeddings of CIFAR-10 dataset extracted by ResNet-18. The reference dog images obtained online and CIFAR-10's own dog images are distributed in the same region. However, the manually pre-defined output representations (PORs) by Zhang *et al.*. in NeuBA [41] fail to cover the dog category.**

Two strategies can be employed to approximate this objective. One is trigger optimization, enhancing the similarity between poisoned and clean images. The other is model optimization, aligning the poisoned embeddings with the reference embeddings.

### 4.2 Transferable Backdoor Attacks

Transferable backdoor attacks aim to embed backdoors into the PTM such that any downstream model built based on it will integrate the backdoor. That is, a backdoored downstream model $\tilde{f}$ behaves normally on clean image $\boldsymbol{x}$, but when the image contains a trigger $\boldsymbol{t}$ specified by the adversary, the model predicts it as the target class $y_t$. We formulate transferable backdoor attacks as the optimization problem:

$$\max_{\theta^{\tilde{F}}} \sum_{(\boldsymbol{x},y) \in \mathcal{D}_t} [\mathbb{I}(\tilde{f}(\boldsymbol{x}) = y) + \mathbb{I}(\tilde{f}(\boldsymbol{x} \oplus \boldsymbol{t}) = y_t)], \quad (2)$$

where $\mathbb{I}(\cdot)$ is the indicator function, which is 1 if $\cdot$ is true and 0 otherwise. Unfortunately, without prior knowledge of the downstream model $f$ and dataset $\mathcal{D}_t$, the adversary cannot solve Eq. 2.

To bridge the gap between the adversary's goals and capabilities, we have enacted two conversions on the above optimization problem. (1) The goal of misclassifying poisoned samples is shifted to producing embedding indistinguishability between poisoned samples and clean samples, *i.e.*, making the PTM predict similar embeddings for both poisoned samples and target class samples. (2) The access to the downstream dataset is converted to publicly available unlabeled shadow dataset and reference images. According to our observations, reference images downloaded from the Internet can replace the real target class images.

### 4.3 Pre- and Post-indistinguishability

To craft a durable and task-agnostic backdoor, we further decomposing the aforementioned embedding indistinguishability into pre- and post-indistinguishability. Pre-indistinguishability represents the indistinguishability between the embeddings generated by the

clean PTM for both the poisoned and clean samples of the target class. We formally define it below.

**Definition 1 (Pre-indistinguishability).** *Let $\boldsymbol{x} \oplus \boldsymbol{t}$ be the poisoned sample and $\boldsymbol{x}_t$ the clean sample of target class. $\boldsymbol{x} \oplus \boldsymbol{t}$ and $\boldsymbol{x}_t$ are deemed pre-indistinguishable if their embeddings, extracted by the clean PTM $F$, exceed the similarity threshold $\epsilon_1$ when measured by $d(\cdot, \cdot)$:*

$$d(F(\boldsymbol{x} \oplus \boldsymbol{t}), F(\boldsymbol{x}_t)) > \epsilon_1. \quad (3)$$

Pre-indistinguishability ensures the durability of backdoor attacks by outlining the inherent similarity between poisoned and clean samples in the embedding space. Since it is independent of the backdoor, it remains unaltered by the fine-tuning process. The most intuitive way to attain pre-indistinguishability is through optimizing the trigger. In this study, we utilize a trigger optimization strategy, which is formally expressed as:

$$\max_{\boldsymbol{t}} \frac{1}{|\mathcal{D}_s| \cdot |\mathcal{D}_r|} \sum_{\boldsymbol{x}_s \in \mathcal{D}_s} \sum_{\boldsymbol{x}_r \in \mathcal{D}_r} d(F(\boldsymbol{x}_s \oplus \boldsymbol{t}), F(\boldsymbol{x}_r)), \quad (4)$$

where $d(\cdot, \cdot)$ measures the similarity (*e.g.*, cosine similarity) between two embeddings. $|\mathcal{D}_s|$ and $|\mathcal{D}_r|$ denote the number of shadow and reference images, respectively. It should be noted that the adversary only needs a few (*i.e.*, $|\mathcal{D}_r| \leq 10$) reference images for target class.

On other hand, post-indistinguishability pertains to the indistinguishability of embeddings from a backdoored PTM, whcih is formally defined as follows.

**Definition 2 (Post-indistinguishability).** *Let $\boldsymbol{x} \oplus \boldsymbol{t}$ represent the poisoned sample and $\boldsymbol{x}_t$ the clean sample of target class. $\boldsymbol{x} \oplus \boldsymbol{t}$ and $\boldsymbol{x}_t$ are post-indistinguishable if their embeddings, extracted by the backdoored PTM $\tilde{F}$, exceed the similarity threshold $\epsilon_2$:*

$$d(\tilde{F}(\boldsymbol{x} \oplus \boldsymbol{t}), \tilde{F}(\boldsymbol{x}_t)) > \epsilon_2. \quad (5)$$

Post-indistinguishability further reinforces the task-agnostic property of backdoor attacks. It implies that the embeddings derived from a backdoored PTM for poisoned samples strongly resemble those of target class samples. This similarity results in predicting the poisoned sample as the corresponding target label when the target class is incorporated in any downstream task, transcending the confines of a single specific task. We formalize the backdoor training as follows:

$$\max_{\theta^{\tilde{F}}} \frac{1}{|\mathcal{D}_s| \cdot |\mathcal{D}_r|} \sum_{\boldsymbol{x}_s \in \mathcal{D}_s} \sum_{\boldsymbol{x}_r \in \mathcal{D}_r} d(\tilde{F}(\boldsymbol{x}_s \oplus \boldsymbol{t}), \tilde{F}(\boldsymbol{x}_r)). \quad (6)$$

The combinations of Eq. 4 and Eq. 6 is a non-convex multi-objective optimization problem, resulting in a durable and task-agnostic backdoor attack.

### 4.4 Two-Stage Optimization

We address pre- and post-indistinguishability with a two-stage optimization.
**Trigger optimization.** Patch-like triggers are extensively utilized in existing work. However, discrete pixel values do not facilitate optimization. One straightforward solution is to adopt global perturbations as the trigger. More specifically, our pervasive triggers

**Figure 3: The pipeline of TransTroj. We first optimize a trigger to make the poisoned images similar to the reference images, *i.e.*, pre-indistinguishability. Then, we optimize the victim PTM such that the poisoned embeddings and reference embeddings cannot be distinguished, *i.e.*, post-indistinguishability.**

are slight perturbations applied to every pixel. The operation of embedding a trigger in an image is as follows:

$$x \oplus t = \text{clip}(x + t, 0, 255), \tag{7}$$

where $\text{clip}(\cdot, 0, 255)$ restricts the pixels of poisoned images to be valid. Note that $x$ and $t$ share the same shape, *e.g.*, $224 \times 224 \times 3$. We achieve a balance between stealthiness and effectiveness by constraining the infinity norm of the trigger, *i.e.*, $\|t\|_\infty \leq \xi$. The objective of trigger optimization can be quantified by the loss function as follows:

$$\mathcal{L}_{\text{pre}} = -\frac{1}{|\mathcal{D}_s|} \sum_{x \in \mathcal{D}_s} d(F(x \oplus t), r), \tag{8}$$

where $r$ is the reference embedding, which is computed as the average of all reference image embeddings, *i.e.*, $r = \frac{1}{n} \sum_{i=1}^{|\mathcal{D}_r|} F(x_{r_i})$. Hence, the optimized trigger $t$ is obtained as a solution to the following optimization problem:

$$\arg \min_t \mathcal{L}_{\text{pre}}, \quad \text{s.t.} \quad \|t\|_\infty \leq \xi. \tag{9}$$

**Victim PTM optimization.** Besides the attack success rate, a proficiently designed backdoor should also uphold the model's original functionality. These two objectives are identified as the effectiveness goal and the functionality-preserving goal, respectively. To fulfill the effectiveness goal, we optimize the victim PTM to produce similar embeddings for poisoned and clean images. We propose an effectiveness loss to formally quantify this objective:

$$\mathcal{L}_{\text{post}} = \frac{1}{|\mathcal{D}_s|} \sum_{x \in \mathcal{D}_s} d(\tilde{F}(x \oplus t), r), \tag{10}$$

To further ensure the model retains its original functionality, it is necessary for the backdoored PTM and the clean PTM to predict similar embeddings for clean samples. We introduce a functionality-preserving loss to quantify this objective:

$$\mathcal{L}_{\text{func}} = \frac{1}{|\mathcal{D}_s|} \sum_{x \in \mathcal{D}_s} d(\tilde{F}(x), F(x)), \tag{11}$$

Thus, the backdoored PTM can be derived from this optimization problem:

$$\arg \min_{\theta^{\tilde{F}}} \mathcal{L} = \mathcal{L}_{\text{post}} + \lambda \mathcal{L}_{\text{func}}, \tag{12}$$

where $\lambda$ is a hyperparameter that balances the effectiveness loss $\mathcal{L}_{\text{post}}$ and functionality-preserving loss $\mathcal{L}_{\text{func}}$. We employ mini-batch gradient descent to solve the optimization problems Eq. 9 and Eq. 12.

## 5 Evaluation

### 5.1 Experimental Setup

To extensively evaluate our TransTroj, we conduct an series of experiments across a diverse range of pre-trained models and downstream tasks.

**Pre-trained models and datasets.** We employ four commonly used PTMs—ResNet [16], VGG [18], ViT [9], and CLIP [27]—as victim models. ResNet and VGG, CNN models, and ViT, a Transformer [34] model, are pre-trained on the ImageNet1K [8] dataset. CLIP, pre-trained on a variety of (*image*, *text*) pairs, contains an image encoder and a text encoder, but our backdoor attacks specifically target the image encoder. To reduce pre-training costs, we utilized pre-trained weights provided by PyTorch and Hugging Face.

**Downstream tasks.** To fully demonstrate the generalization of our backdoor attack, we select six downstream tasks, including CIFAR-10 [19], CIFAR-100 [19], GTSRB [32], Caltech 101 [11], Caltech 256 [13] and Oxford-IIIT Pet [26]. More details can be found in Appendix A.

**Evaluation metrics.** Following Jia *et al.* [17], our evaluation employs three key measurement metrics. *Clean Accuracy (CA)* refers to the classification accuracy of the clean downstream model, serving as the baseline performance for the downstream task. *Attack Success Rate (ASR)* denotes the proportion in which the backdoored downstream model classifies poisoned samples as the target class. *Backdoored Accuracy (BA)* signifies the classification accuracy of the backdoored downstream model, quantifying the performance of the backdoored model on its benign task. Comparing BA with CA allows us to determine whether the backdoor maintains the original functionality of the victim PTM.

**Implementation settings.** We download 10 reference images for each target class from the Internet. The shadow dataset $\mathcal{D}_s$ consists of 50,000 images randomly sampled from ImageNet1K. It's important to note that the labels of these shadow images are not utilized. The infinity norm of the optimized trigger is constrained,

 

**Table 2: Comparison of attack performance on different PTMs and downstream tasks. We conducted backdoor attack experiments on three different pre-trained image encoders and six different downstream tasks. All values are percentages.**

| Method | Model | CIFAR-10 | | | CIFAR-100 | | | GTSRB | | | Caltech 101 | | | Caltech 256 | | | Oxford-IIIT Pet | | |
|---|---|---|---|---|---|---|---|---|---|---|---|---|---|---|---|---|---|---|---|
| | | CA | BA | ASR | CA | BA | ASR | CA | BA | ASR | CA | BA | ASR | CA | BA | ASR | CA | BA | ASR |
| BadEncoder | ResNet-18 | 95.45 | 95.20 | 9.81 | 80.11 | 79.48 | 1.12 | 98.54 | 98.84 | 5.75 | 96.14 | 95.68 | 1.04 | 82.03 | 81.49 | 26.87 | 88.09 | 88.50 | 81.28 |
| | VGG-11 | 91.52 | 91.18 | 9.64 | 70.67 | 70.93 | 3.81 | 99.08 | 99.02 | 5.71 | 93.09 | 93.95 | 58.99 | 74.64 | 74.50 | 33.59 | 87.03 | 83.73 | 75.14 |
| | ViT-B/16 | 98.26 | 97.85 | 0.09 | 85.78 | 86.33 | 0.41 | 98.65 | 98.87 | 0.25 | 96.72 | 96.03 | 0.06 | 85.47 | 85.70 | 0.07 | 92.75 | 92.64 | 0.11 |
| NeuBA | ResNet-18 | 92.07 | 91.02 | 13.74 | 73.33 | 71.67 | 4.62 | 95.60 | 95.02 | 7.71 | 88.13 | 85.54 | 9.85 | 62.56 | 58.38 | 2.92 | 60.43 | 48.98 | 4.47 |
| | VGG-11 | 91.93 | 90.94 | 57.17 | 70.56 | 70.88 | 49.35 | 96.28 | 95.86 | 53.29 | 89.69 | 89.69 | 63.88 | 62.22 | 59.74 | 48.69 | 69.58 | 69.69 | 60.53 |
| | ViT-B/16 | 95.95 | 96.18 | 81.95 | 84.49 | 84.65 | 78.19 | 95.74 | 95.81 | 67.91 | 88.88 | 88.94 | 64.92 | 75.72 | 75.37 | 56.30 | 73.48 | 74.05 | 64.98 |
| Ours | ResNet-18 | 95.45 | 95.41 | **100.0** | 80.11 | 80.25 | **100.0** | 98.54 | 98.80 | **100.0** | 96.14 | 95.79 | **98.68** | 82.03 | 81.27 | **98.79** | 88.09 | 88.42 | **99.73** |
| | VGG-11 | 91.52 | 92.00 | **99.51** | 70.67 | 71.49 | **100.0** | 99.08 | 99.13 | **94.03** | 93.09 | 95.45 | **93.95** | 74.64 | 74.37 | **91.47** | 87.03 | 83.46 | **99.07** |
| | ViT-B/16 | 98.26 | 97.91 | **100.0** | 85.78 | 86.03 | **100.0** | 98.65 | 98.95 | **100.0** | 96.72 | 96.08 | **99.36** | 85.47 | 85.55 | **99.80** | 92.75 | 89.26 | **100.0** |

such that $\|t\|_\infty \leq \xi = 10$. We set the ratio $\lambda$, which represents the relationship between $\mathcal{L}_{post}$ and $\mathcal{L}_{func}$, to 10. The fine-tuning learning rate is 1e-4 for ResNet and VGG, and 1e-5 for ViT and CLIP. We observe that the model converges to satisfactory performance within a few epochs. To evaluate whether the backdoor is durable, we fine-tune 20 epochs for all downstream tasks.

**Baseline methods.** We compare our method against two SOTA attacks, BadEncoder [17] and NeuBA [41], both specifically designed for attacking PTMs. We utilize the publicly available implementations provided by the authors of BadEncoder and NeuBA. Additionally, we extend NeuBA from binary classification tasks to support downstream multi-class classification.

## 5.2 Attack Effectiveness

**Comparison to existing attacks.** We evaluate the attack performance on various PTMs and downstream tasks. For each task, we adhere to the system model delineated in Sec. 3.1 to fine-tune the downstream model. Tab. 2 shows the results. It is evident that TransTroj achieves significantly higher ASRs than either BadEncoder or NeuBA across different PTMs and tasks. Notably, BadEncoder records ASRs below 10% on most tasks. Contrarily, TransTroj garners high ASRs, exceeding 99% on all tasks when employing ViT-B/16. The least successful case is Caltech 256 on VGG-11, which still achieves a 91.47% ASR. We attribute this to potentially small inter-class distances in the embedding space.

**Functionality-preserving.** Our method effectively maintains the functionality of the backdoored PTM. Observations from Tab. 2 indicate that, for most downstream tasks, the deviation between the backdoor accuracies and the clean accuracies are within 1%. In some instances, the backdoor accuracies even surpass the clean accuracies. Overall, these results suggest that downstream models, even when fine-tuned from the backdoored PTM, can still preserve the core functionality for downstream tasks. Consequently, identifying the backdoor by solely insecting the performance of downstream tasks presents a significant challenge.

**Durable.** Our backdoor maintains its effectiveness throughout the fine-tuning process. We record the ASR after each epoch of fine-tuning, as illustrated in Fig. 4. It is noticeable that our method demonstrates a remarkably stable ASR during the fine-tuning process, with fluctuations scarcely exceeding 1%. The least successful case is the Clatech 101 task, where the ASR commences at 99.71%

**Table 3: Results of simultaneously attacking three target downstream datasets through one target class. $\uparrow_{0.14}$ indicates that the BA is 0.14% higher than the CA.**

| Target class | Downstream dataset | CA | BA | ASR |
|---|---|---|---|---|
| Sunflower | CIFAR-100 | 80.11 | $80.25_{\uparrow 0.14}$ | 100.0 |
| | Caltech 101 | 96.14 | $95.79_{\downarrow 0.35}$ | 98.68 |
| | Caltech 256 | 82.03 | $81.27_{\downarrow 0.76}$ | 98.79 |
| Leopard | CIFAR-100 | 80.11 | $80.17_{\uparrow 0.06}$ | 100.0 |
| | Caltech 101 | 96.14 | $95.85_{\downarrow 0.29}$ | 97.24 |
| | Caltech 256 | 82.03 | $81.29_{\downarrow 0.74}$ | 99.04 |
| Kangaroo | CIFAR-100 | 80.11 | $79.71_{\downarrow 0.40}$ | 99.99 |
| | Caltech 101 | 96.14 | $95.68_{\downarrow 0.46}$ | 98.21 |
| | Caltech 256 | 82.03 | $81.32_{\downarrow 0.71}$ | 97.39 |

after the first epoch and diminishes to 98.68% after 20 epochs, a decline of 1.02%. In contrast, the ASRs of the baselines display instability. For instance, NeuBA's accuracy on the Caltech 256 dramatically decreases from 97.33% in the second epoch to only 46.32% in the 12th epoch, a substantial drop of 51.01%. All these results suggest that the backdoors injected using our method are durable, maintaining their effectiveness even after fine-tuning.

**Task-agnostic.** Our method is capable of attacking multiple downstream tasks using a single target class. Specifically, we select three target classes (Sunflower, Leopard and Kangaroo) that concurrently exist in the CIFAR-100, Caltech 101, and Caltech 256 datasets. As depicted in Tab. 3, the backdoor can be successfully activated across these three downstream tasks. For instance, when the target class is "Sunflower", the attack success rates for CIFAR-100, Caltech 101, and Caltech 256 reach 100%, 98.68%, and 98.79%, respectively. These results indicate that our method is task-agnostic, implying that the backdoor can be effectively activated as long as the target class is included in the downstream task.

## 5.3 Multi-target Backdoor Attacks

While all previous experiments have focused on attacking a single target class, this section evaluates the effectiveness of our method in simultaneously attacking multiple target classes. Specifically,


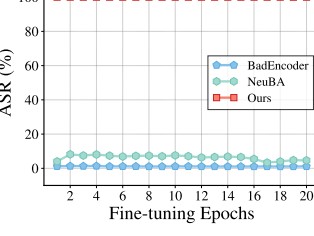
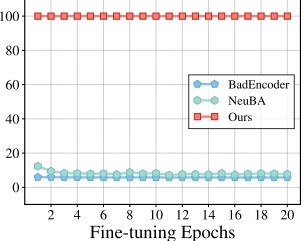
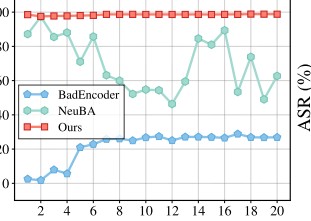
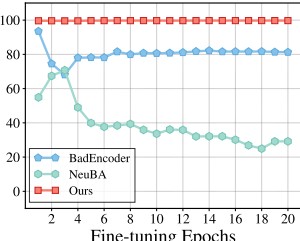

**Figure 4: Attack success rates when fine-tuning the backdoored PTM for different downstream tasks. BadEncoder and NeuBA achieve only limited performance across a subset of downstream tasks. In contrast, our method achieves a high attack success rate across various downstream tasks and remains stable during the fine-tuning process.**

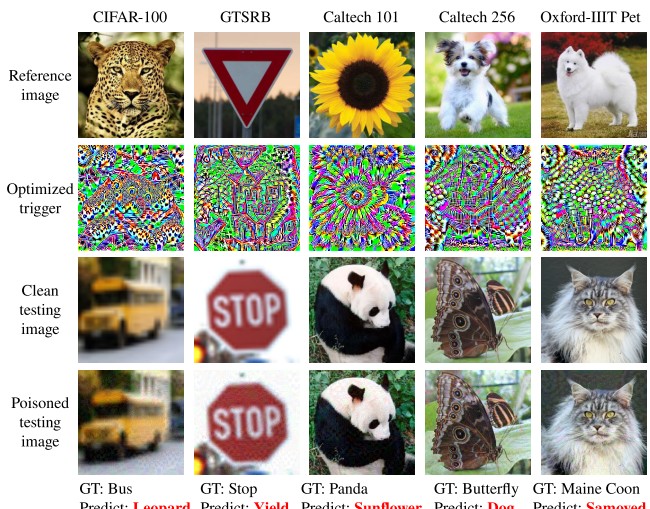

**Figure 5: Visualization of the multi-target backdoor attack. The downstream models fine-tuned based on the backdoored PTM predict the poisoned samples as the class labels corresponding to the reference images, rather than the ground truth (GT) labels.**

**Table 4: Results of attacking 5 target classes simultaneously.**

| Downstream dataset | CA | BA | ASR |
|---|---|---|---|
| CIFAR-100$_{Leopard}$ | 80.11 | 79.75$\downarrow_{0.36}$ | 99.89 |
| GTSRB$_{Yield\ sign}$ | 98.54 | 98.80$\uparrow_{0.26}$ | 99.98 |
| Caltech 101$_{Sunflower}$ | 96.14 | 95.45$\downarrow_{0.69}$ | 98.21 |
| Caltech 256$_{Dog}$ | 82.03 | 81.04$\downarrow_{0.99}$ | 99.27 |
| Oxford-IIIT Pet$_{Samoyed}$ | 88.09 | 87.79$\downarrow_{0.30}$ | 99.51 |

the adversary selects multiple target classes $(y_1, y_2, ..., y_n)$ and optimizes corresponding triggers $(t_1, t_2, ..., t_n)$. Following this, he/she trains the victim PTM to bind each target class $y_i, 1 \leq i \leq n$ with its respective trigger $t_i, 1 \leq i \leq n$. To evaluate our TransTroj in such a scenario, as depicted in Fig. 5, we use ResNet-18 as the victim model and simultaneously attack five downstream tasks (*i.e.*, CIFAR-100, GTSRB, Caltech 101, Caltech 256 and Oxford-IIIT Pet) with different

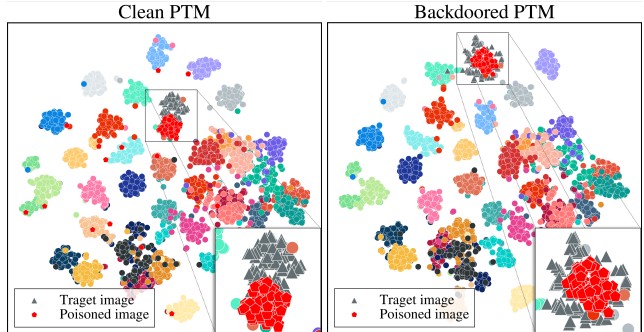

**Figure 6: Visualization of dimension-reduced embeddings using ResNet-18 as the PTM and Oxford-IIIT Pet as the downstream task. Differently colored dots denote clean images of various classes; triangles mark the target class, and pentagons represent randomly selected poisoned images from the test set.**

target classes (*i.e.*, Leopard, Yield, Sunflower, Dog and Samoyed). It is important to emphasize that we trained only one backdoor model and fine-tuned it for five different downstream tasks, rather than training separate backdoor models for each task. Detailed results in Tab. 4 indicate that our attacks can achieve high ASRs (exceeding 99%) when targeting multiple calsses simultaneously, while still maintaining accuracy of the downstream models.

## 5.4 Cause Analysis

Our TransTroj's high ASR is affirmed by both theoretical and empirical analyses. Theoretically, as discussed in Sec. 4.3, the key to the success of our backdoor attack lies in the indistinguishability between poisoned samples and target class samples within the embedding space. Fig. 6 (left) empirically supports this, revealing pre-indistinguishability due to the optimized trigger. This indistinguishability becomes absolute in the feature space following backdoor training, as evidenced by Fig. 6 (right). Further data analysis, as shown in Fig. 7, demonstrates that the backdoored PTM's attention is predominantly centered on the poisoned image's middle area. Thus, the downstream model ignoring the specific content of the poisoned input, predicting it as the target class.


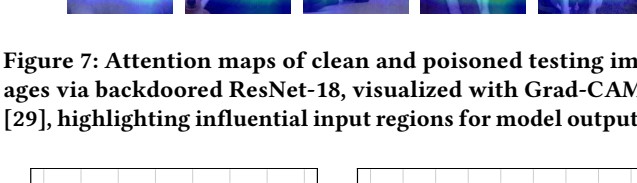
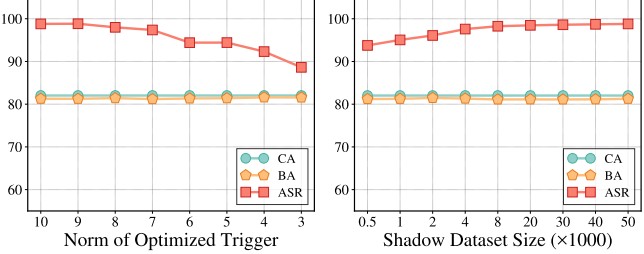

Figure 7: Attention maps of clean and poisoned testing images via backdoored ResNet-18, visualized with Grad-CAM [29], highlighting influential input regions for model output.

Figure 8: The impact of the optimized trigger infinity norm $\xi$ (left) and the shadow dataset size $|\mathcal{D}_s|$ (right).

## 5.5 Sensitivity Analysis

We investigate various factors, such as trigger infinity norm and shadow dataset size, that may impact the performance of TransTroj. For these studies, ResNet-18 is chosen as the PTM, Caltech 256 serves as the downstream task, and "Sunflower" is selected as the target class.

**Trigger infinity norm.** Gradually reduce the infinity norm $\xi$ of triggers from 10 to 3, we evaluate the attack performance on Caltech 256, as shown in Fig. 8 (left). It's obvious that the attack success rate significantly decreases with the reduction of the infinity norm, due to the failure of trigger optimization to make the poisoned embeddings resemble the clean embeddings when the infinity norm is excessively small. In other words, pre-indistinguishability cannot be ensured.

**Shadow dastaset size.** Subsets of different sizes (ranging from 512 to 50,000) are randomly sampled from ImageNet1K to serve as shadow datasets. As shown in Fig. 8 (right), our method achieves high attack success rates and preserves accuracy of the downstream model once the shadow dataset size surpasses 10,000. Notably, the ImageNet1K training set contains over one million images, thus indicating that the collection of the shadow dataset is not challenging for the adversary.

**Ablation study.** We conduct a comprehensive ablation study to evaluate the effectiveness of our method, including experiments on the impact of different *loss terms*, *trigger patterns*, and *the use of reference images*. For more detailed analysis and results, please refer to Appendix C.

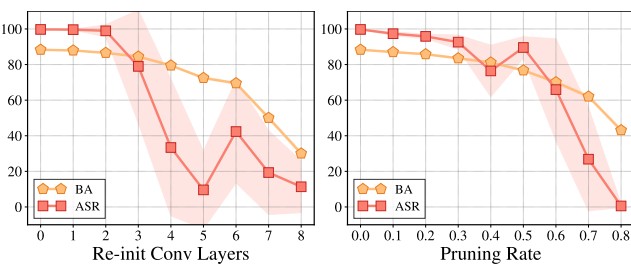

Figure 9: The average (compiled from 100 trials) attack success rate and backdoor accuracy after re-initialization and fine-pruning. Note that ResNet-18 consists of 17 convolutional layers and one fully connected layer.

## 5.6 Robustness Against Defenses

This section investigates the robustness of TransTroj against model reconstruction based backdoor defenses, specifically re-initialization and fine-pruning. We employ ResNet-18 as the PTM, designate Oxford-IIIT Pet as the downstream task, and select "Samoyed" as the target class.

**Re-initialization.** An intuitive strategy to resist backdoor is the re-initialization of the final few layers of PTMs. We re-initialized the last $n, 0 \leq n \leq 8$ convolutional layers of ResNet-18 before fine-tuning. As depicted in Fig. 9 (left), model accuracy diminishes with an increase in the number of re-initialized layers, while the backdoor sustains a high SAR until re-initialization extends to the last four layers. Subsequently, the ASR declines from 99.73% to 32.92%. Simultaneously, the model accuracy also decreases from 88.33% to 79.49%. This indicates that re-initialization cannot balance between model utility and backdoor defense, providing evidence of TransTroj's robustness against re-initialization.

**Fine-pruning.** Fine-pruning [22] aims to erase backdoor by deactivating neurons that are dormant on clean inputs. We employed a mask to block inactive channels following each residual block in ResNet-18. The proportion of channels that are masked is controlled by the pruning rate. As depicted in Fig. 9 (right), even with 50% of the channels are pruned, the ASR can still reach 89.59%, but the model accuracy has dropped from 88.33% to 76.73%. Further pruning leads to degradation in both model performance and backdoor effectiveness. Thus, fine-pruning proves to be ineffective in defending our TransTroj.

## 6 Conclusion

In this paper, we have introduced **TransTroj**, a novel backdoor attack that effectively compromises pre-trained models (PTMs) by exploiting embedding indistinguishability. By formalizing the backdoor insertion as an indistinguishability problem between poisoned and clean samples in the embedding space, we address the critical challenges of durability and task-agnosticism that limit existing backdoor attacks on PTMs. Extensive experiments conducted on four widely used PTMs – ResNet, VGG, ViT, and CLIP – and six downstream tasks demonstrate that TransTroj significantly outperforms SOTA task-agnostic backdoor attacks.

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

# A Details of Downstream Tasks

We utilize six different downstream datasets in our experiments. Below, we provide details for each dataset:

**CIFAR-10.** [19]: This dataset consists of 60,000 low-resolution images (32×32 pixels) divided into 10 classes, with 50,000 images for training and 10,000 for testing. It features common objects such as airplanes, automobiles, and birds.

**CIFAR-100.** [19]: Similar to CIFAR-10 but more fine-grained, this dataset contains 60,000 images across 100 classes, with the same training and testing split of 50,000 and 10,000 images, respectively.

**GTSRB.** [32]: The German Traffic Sign Recognition Benchmark includes 51,800 images of traffic signs categorized into 43 classes. It provides 39,200 images for training and 12,600 for testing.

**Caltech 101.** [11]: This dataset comprises 8,677 images of objects from 101 categories. We randomly select 80% of the images for training and use the remaining 20% for testing.

**Caltech 256.** [13]: An extension of Caltech 101, it contains 29,780 images spanning 256 object categories. We apply the same random split of 80% for training and 20% for testing.

**Oxford-IIIT Pet.** [26]: This dataset features 7,349 images of 37 breeds of cats and dogs. It is divided into 3,680 images for training and 3,669 for testing.

# B Implementation Details

The attack chain involves three stages: pre-training, backdoor injection, and fine-tuning. The following provides implementation details for each stage.

**Pre-training.** In our experiments, we use four different PTMs (*i.e.* ResNet [16], VGG [18], ViT [9], and CLIP [27]). ResNet, VGG, and ViT are pre-trained on the ImageNet1K dataset using supervised learning methods. To reduce pre-training costs, we download the pre-trained weights from PyTorch. CLIP is pre-trained on a variety of (*image, text*) pairs using self-supervised learning technique. We download it from Hugging Face.

**Backdoor injection.** The backdoor is injected into a PTM. The dataset used during backdoor injection is called Shadow Dataset. By default, the shadow dataset contains 50,000 images randomly sampled from the ImageNet1K training dataset. To obtain reference embeddings, we download 10 reference images for each target class from the Internet, as shown in Fig. 10. Note that different PTMs and target classes result in different optimized triggers, as shown in Fig. 11. The infinity norm constraint on the trigger affects the stealthiness of the optimized trigger. When the infinity norm is small, the optimized trigger is imperceptible, as shown in Fig. 12. Unless stated otherwise, we set the infinity norm constraint of the trigger to 10. All experiments are conducted on a server equipped with eight NVIDIA RTX 3090 GPU and 3.5GHz Intel CPUs.

**Fine-tuning.** The dataset used in fine-tuning the downstream model is referred as the downstream dataset. In our experiments, different datasets (*i.e.*, CIFAR-10, CIFAR-100, GTSRB, Caltech 101, Caltech 256, and Oxford-IIIT Pet) are used as downstream datasets. The training of the downstream model uses the cross-entropy loss function and Adam optimizer. When the PTM is ResNet-18 or VGG11, the learning rate is set to 1e-4. When the PTM is ViT-B/16 or CLIP, the learning rate is set to 1e-5. Note that all downstream tasks are fine-tuned for 20 epochs.

**Table 5: Results of attacking "Sunflower" using various trigger patterns. Fig. 14 displays these four types of triggers.**

| Tigger pattern | Downstream dataset | CA | BA | ASR |
|---|---|---|---|---|
| Patch | CIFAR-100 | 80.11 | 79.85$\downarrow$ 0.26 | 68.84 |
| | Caltech 101 | 96.14 | 95.74$\downarrow$ 0.40 | 10.94 |
| | Caltech 256 | 82.03 | 81.34$\downarrow$ 0.69 | 6.29 |
| SIG | CIFAR-100 | 80.11 | 79.52$\downarrow$ 0.59 | 1.43 |
| | Caltech 101 | 96.14 | 95.74$\downarrow$ 0.40 | 90.61 |
| | Caltech 256 | 82.03 | 81.79$\downarrow$ 0.24 | 74.69 |
| Random | CIFAR-100 | 80.11 | 79.44$\downarrow$ 0.67 | 1.54 |
| | Caltech 101 | 96.14 | 95.51$\downarrow$ 0.37 | 19.64 |
| | Caltech 256 | 82.03 | 81.36$\downarrow$ 0.67 | 0.84 |
| Optimized | CIFAR-100 | 80.11 | 80.25$\uparrow$ 0.14 | 100.0 |
| | Caltech 101 | 96.14 | 95.79$\downarrow$ 0.35 | 98.68 |
| | Caltech 256 | 82.03 | 81.27$\downarrow$ 0.76 | 98.79 |

# C Ablation Study

We study the effect of loss terms, trigger pattern and reference images. For all experiments here, the PTM is ResNet.

**Loss terms.** Our TransTroj incorporates three loss terms, *i.e.* $\mathcal{L}_{pre}$, $\mathcal{L}_{post}$ and $\mathcal{L}_{func}$. We also include a hyperparameter $\lambda$ to balance $\mathcal{L}_{post}$ and $\mathcal{L}_{func}$. Therefore, it is important to examine the impact of $\lambda$ on our method. The results of this examination are displayed in Fig. 13. We observe that both an exceedingly large or samll $\lambda$ prevents TransTroj from achieving a high attack success rate while preserving accuracy. In particular, the model accuracy starts to decrease when $\lambda$ exceeds approximately 10. Therefore, both $\mathcal{L}_{post}$ and $\mathcal{L}_{func}$ are important for effective backdoor training.

**Trigger pattern.** To demonstrate the necessity of the trigger optimization, we replaced our optimized triggers with different trigger patterns while keeping all other settings unchanged. Fig. 14 displays the various triggers. The experimental results indicate that neither patch-like triggers nor pervasive triggers achieve high ASRs. This is because these triggers do not make the poisoned embeddings similar to the clean embeddings of the target class.

**Single reference image.** In the aforementioned experiments, we use the average embedding of 10 reference images as the reference embedding. Here, we study whether the reference embedding could be a single image embedding. We utilize the embeddings from a single "Yield sign" image and a single "Dog" image to attack GTSRB and CIFAR-10, respectively. As shown in Fig. 15, we find that some reference images can cause the backdoor attack to fail. For instance, using the embedding of "Dog$_5$" as the reference embedding only achieved 6.43% attack success rate. This is because the downstream model is unable to correctly classify "Dog$_5$". According to Fig. 2b, not all reference images can be correctly classified by the downstream model. Hence, employing multiple reference images is an effective strategy to avoid such risk.

# D Other Transfer Methods

In addition to fine-tuning, PTMs have more application scenarios, such as linear probing and zero-shot classification. Linear probing utilizes the PTM to project inputs to an embedding space, and then

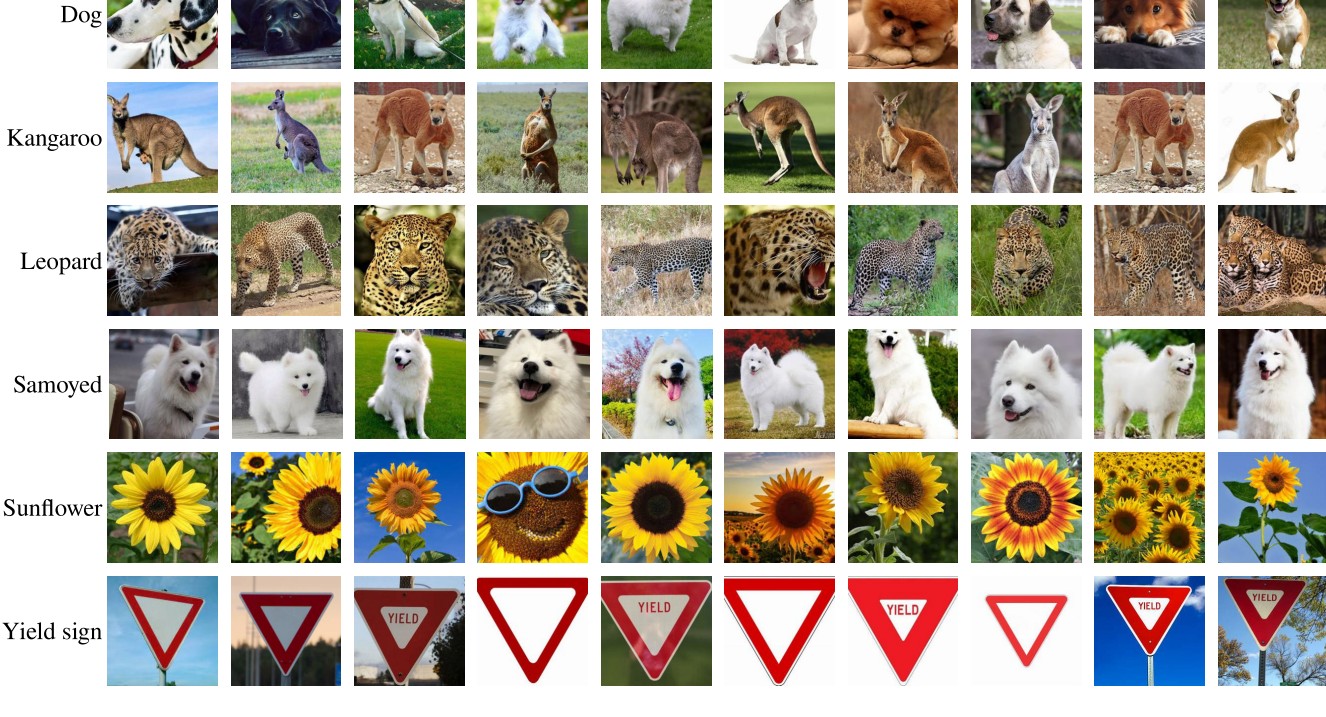

**Figure 10: Visualization of reference iamges. We download 10 reference images for each target category from the Internet.**

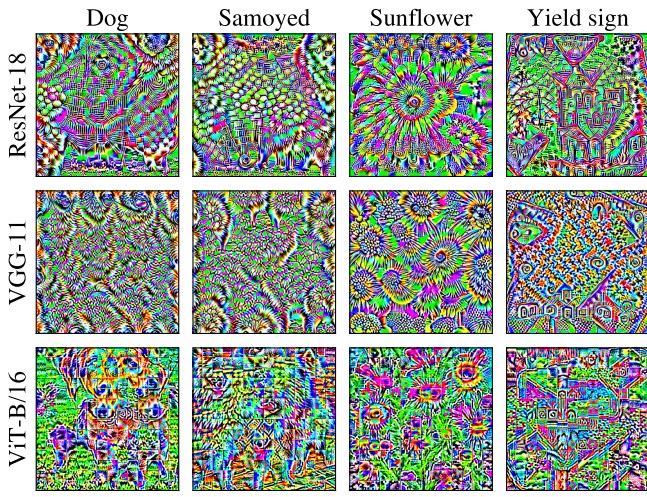

**Figure 11: Visualization of optimized triggers. Different PTMs and target classes lead to different triggers.**

trains a linear classifier to map embeddings to downstream classification labels. Zero-shot classification trains an image encoder and a text encoder that map images and texts to the same embedding space. The similarity of the two embeddings from an image and a piece of text is used for prediction.

To verify the effectiveness of TransTroj in other application scenarios, we use CLIP as the victim model. CLIP consists of both

**Table 6: Comparison of three different transfer scenarios. TransTroj achieves high attack success rates and maintains the accuracy of the downstream tasks when attacking CLIP.**

| Transfer | Downstream Dataset | CA | BA | ASR |
|---|---|---|---|---|
| Zero-shot | CIFAR-100 | 61.87 | 59.96 | 100.0 |
| | Caltech 101 | 84.04 | 83.99 | 100.0 |
| | Caltech 256 | 85.32 | 83.52 | 100.0 |
| Linear probing | CIFAR-100 | 78.48 | 78.30 | 100.0 |
| | Caltech 101 | 94.76 | 95.12 | 100.0 |
| | Caltech 256 | 90.19 | 89.74 | 100.0 |
| Fine-tuning | CIFAR-100 | 83.88 | 82.52 | 100.0 |
| | Caltech 101 | 95.97 | 95.45 | 98.85 |
| | Caltech 256 | 88.54 | 86.84 | 99.11 |

an image encoder and a text encoder. We apply TransTroj to inject a backdoor to the image encoder. In the zero-shot classification scenario, we adopt sentence "A photo of a class name" as the context sentence. In the linear probing scenario, we use a fully connected neural network with two hidden layers as the downstream classifier. The training of the downstream classifier uses the cross-entropy loss function and Adam optimizer. It takes 500 epochs with an initial learning rate of 0.0001.

Tab. 6 shows the experimental results. We find that our TransTroj achieves high attack success rates and maintains the accuracy of the downstream tasks in all three transfer scenarios. Our experimental

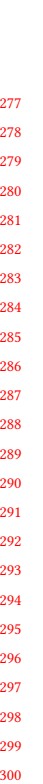

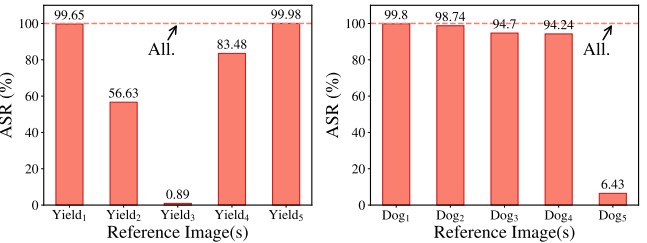

$\xi$=10  $\xi$=9  $\xi$=8  $\xi$=7  $\xi$=6  $\xi$=5  $\xi$=4  $\xi$=3

**Figure 12: Visual effects of optimized triggers under different infinite norm constraints. Note that we scale the optimized triggers to $[0, 255]$ using min-max normalization for visualization. Row 1: optimized triggers. Row 2: poisoned samples.**

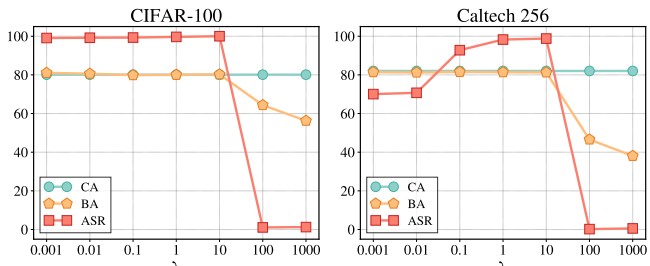

**Figure 15: Results of attacking "Yield" and "Dog" using a single reference image. "All" refers to the attack success rate when using the average embedding of 10 reference images.**

**Figure 13: The impact of the $\lambda$.**

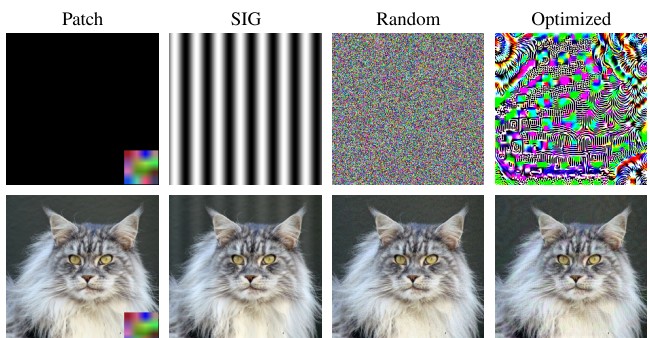

**Figure 14: Comparison of different triggers. The patch trigger is generated according to [28] and has a size of 50×50. The SIG trigger is generated by the horizontal sinusoidal function defined in [1] with $\Delta = 10$ and $f = 32$. The random trigger is sampled from a uniform distribution between [-5, 5]. The optimized trigger is produced by our trigger optimization method, with the target class "Samoyed" and $\xi = 5$.**

results indicate that our TransTroj is effective when applied to an image encoder pre-trained on a large amount of (*image*, *text*) pairs. Note that zero-shot classification and linear probing can only achieve suboptimal performance on some downstream tasks. This further demonstrates the necessity of fine-tuning.

