# OpenReview forum: "Model Supply Chain Poisoning: Backdooring Pre-trained Models via Embedding Indistinguishability"
_ACM.org/TheWebConf/2025/Conference — WWW 2025 Oral_

### Official Review · Reviewer_Dgbf · 2024-11-11

**Novelty:** 6
**Technical Quality:** 6

**Review:**

## Summary:
This paper proposes a transferable backdoor attack, called TransTroj, on pre-trained models. This attack is task-agnostic, enabling it to propagate across different downstream tasks in the model supply chain.  In detail, to ensure that the embeddings of poisoned samples and the target class are indistinguishable, it introduces a two-stage optimization that separately optimizes triggers and the model’s parameters. Experiments demonstrate the effectiveness of TransTroj under various system settings.
## Strengths:
1. The proposed attack can transfer to different downstream tasks, which is severe and should be taken seriously.
2. This paper conducts sufficient experiments, demonstrating the superior performance over baseline methods.
3. The paper is well-written and easy to follow.
## Weaknesses:
1. The authors claim that the success rate of backdoor defense techniques depends on the qualitative similarity between the shadow dataset and the dataset used to compute the attack success rate. However, to make this analysis more robust, it would be beneficial to move beyond qualitative assessments and incorporate quantitative measures of similarity. I think the authors could utilize pre-trained encoders such as CLIP to quantify the similarity between datasets.
2. Could you explain the difference between the proposed backdoor attack with the universal adversarial attack [a]? Although backdoor attacks and adversarial attacks have significant differences in their threat models, the global perturbation trigger proposed in this paper seems to have similarities with UAP (Universal Adversarial Perturbations). More insights should be provided.
3. When fine-tuning PTMs for different downstream tasks, the number of fine-tuning steps required may vary depending on the specific dataset. In other words, different tasks may require different numbers of fine-tuning epochs. However, in the experiments, the authors fixed this number at 20 epochs. The authors should explain the rationale behind this experimental setup.
4. While embedding indistinguishability shows promising attack results, the paper lacks in-depth discussion on how it addresses previous issues. In other words, the paper should include more discussion to enhance the interpretability of TransTroj.
[a] Moosavi-Dezfooli S M, Fawzi A, Fawzi O, et al. Universal adversarial perturbations. CVPR, 2017.

**Questions:**

1. How to construct shadow dataset $D_s$?
2. In Table 1, why are the CA results same between BadEncoder and proposed method?
3. Other questions mentioned in the Review-Weaknesses.

**Reviewer Confidence:**

4: The reviewer is certain that the evaluation is correct and very familiar with the relevant literature

**Scope:**

4: The work is relevant to the Web and to the track, and is of broad interest to the community

---

### Official Review · Reviewer_LcRh · 2024-11-25

**Novelty:** 6
**Technical Quality:** 6

**Review:**

## Summary:
The authors proposed a backdoor attack method, TransTroj, targeting pre-trained models in the model supply chain. The proposed method embeds backdoors into PTMs by ensuring that poisoned samples are indistinguishable from clean samples of a target class in the embedding space of the model. Experimental results demonstrate that TransTroj achieves high attack success rates across multiple models and tasks while preserving model accuracy on clean data.
## Strengths:
1.This paper studies an interesting setting: backdooring pre-trained models and transferring the backdoor to downstream tasks.
2.The pre- and post-indistinguishability is very impressive and makes sense.
3. The evaluation is comprehensive, including various types of models, datasets, and an ablation study.
## Weaknesses:
1. The table shows that ASRs for BadEncoder and NeuBA are unexpectedly low across tasks, despite high ASRs reported in their original papers. The reason for this discrepancy isn’t thoroughly discussed in this experimental setting.
2. The definition of task-agnostic scenarios is unclear, particularly concerning whether the targeted label is considered prior knowledge. Clarification is needed on the role of targeted labels in backdoor attacks against PTMs.
3. The limitations and possible future development of the paper are not discussed, thus leaving little space for the reader to understand how the proposed research can be expanded to advance the SOTA.
4. Minor issues:
- Line 257: “he/her” should be “he/she”
- Line 432: “whcih” should be “which”
- Line 1130: “samll” should be “small”

**Questions:**

1. Why 20 epochs were chosen for the experiments? More reasoning here would be appreciated, and it may help to plot the benign performance in addition to the ASR.

**Reviewer Confidence:**

4: The reviewer is certain that the evaluation is correct and very familiar with the relevant literature

**Scope:**

4: The work is relevant to the Web and to the track, and is of broad interest to the community

---

### Official Review · Reviewer_1Mrj · 2024-11-25

**Novelty:** 4
**Technical Quality:** 5

**Review:**

This paper explores backdoor attacks in the pre-training-then-fine-tuning paradigm. The authors introduce TransTroj, a new backdoor attack designed to implant functionality-preserving, durable, and task-agnostic backdoors in pre-trained models (PTMs) that can seamlessly transfer to downstream fine-tuned models. The attack is formalized as an indistinguishability problem, with a two-stage optimization framework that separately optimizes triggers and target PTMs to achieve embedding indistinguishability. TransTroj is evaluated across four PTMs and six downstream tasks, demonstrating its effectiveness.

Strengths:

- Important topic
- Innovative framework
- Good performance

Weaknesses:

- The baseline attack results in this paper show a significant discrepancy compared to the original study
- Lack of evaluation on baseline attacks and SOTA defenses

**Questions:**

This paper is well-organized and easy to understand. The authors focus on backdooring pre-trained models (PTMs), which is an important topic in the field of backdoor attacks. They propose TransTroj, an innovative attack framework that optimizes both triggers and target models in a two-step process. This approach intuitively appears more effective and stealthier than traditional methods that directly backdoor target models. The experimental results further validate the effectiveness of TransTroj.

My main concern with this study lies in the discrepancy between the baseline backdoor attack results reported in this paper and those reported in the original paper, particularly for BadEncoder. In Table 2, the ASR of BadEncoder on CIFAR-10, CIFAR-100, and GTSRB are below 10%. However, in the original paper, BadEncoder achieves an ASR of over 90% on CIFAR-10 using the same ResNet-18 architecture. A possible explanation for this discrepancy could be differences in the configuration of downstream tasks between the two studies. However, the authors have not provided sufficient details about the downstream models. I recommend that the authors carefully review their experimental code and include fine-tuning details for the downstream model, such as the architecture of the downstream classifier.

In addition, this study lacks comprehensive comparisons with baseline backdoor attacks. I recommend that the authors compare TransTroj to representative works, such as [20, 21, 38, a, b, c], to better position their approach in the context of existing literature. Furthermore, the authors only assess TransTroj against two basic defense methods, re-initialization and fine-pruning. For a new backdoor attack, it is crucial to evaluate its robustness against state-of-the-art defenses, such as STRIP [d], DECREE [e], and [f]. Including these comparisons would strengthen the study and provide a more thorough understanding of the proposed method’s effectiveness.

Minor:

- whcih → which
- an series of → a series of

[a] Saha, Aniruddha, et al. "Backdoor attacks on self-supervised learning." Proceedings of the IEEE/CVF Conference on Computer Vision and Pattern Recognition. 2022.
[b] Li, Changjiang, et al. "An embarrassingly simple backdoor attack on self-supervised learning." Proceedings of the IEEE/CVF International Conference on Computer Vision. 2023.
[c] Wang, Qiannan, et al. "GhostEncoder: Stealthy backdoor attacks with dynamic triggers to pre-trained encoders in self-supervised learning." Computers & Security 142 (2024): 103855.
[d] Gao Y, Xu C, Wang D, et al. Strip: A defence against trojan attacks on deep neural networks[C]//Proceedings of the 35th annual computer security applications conference. 2019: 113-125.
[e] Feng, Shiwei, et al. "Detecting backdoors in pre-trained encoders." Proceedings of the IEEE/CVF Conference on Computer Vision and Pattern Recognition. 2023.
[f] R. Bie, J. Jiang, H. Xie, Y. Guo, Y. Miao and X. Jia, "Mitigating Backdoor Attacks in Pre-Trained Encoders via Self-Supervised Knowledge Distillation," in IEEE Transactions on Services Computing, vol. 17, no. 5, pp. 2613-2625, Sept.-Oct. 2024, doi: 10.1109/TSC.2024.3417279.

**Reviewer Confidence:**

3: The reviewer is confident but not certain that the evaluation is correct

**Scope:**

3: The work is somewhat relevant to the Web and to the track, and is of narrow interest to a sub-community

---

### Official Review · Reviewer_KKZH · 2024-12-05

**Novelty:** 5
**Technical Quality:** 5

**Review:**

This paper proposes a backdoor attack method on pre-trained models through embedding indistinguishability. The attack aligns the backdoor trigger with a pre-defined reference embedding, formalizing the attack as an embedding indistinguishability problem between poisoned and clean samples. It introduces a two-stage optimization approach to implement this idea. Experimental results show that the proposed method effectively addresses two challenges faced by existing works: durability and partial task-agnosticism. Compared to other
baselines, the method demonstrates significantly better performance.

**Questions:**

1. If the optimized trigger corresponds to a target label that does not appear in the downstream task's dataset, can this trigger still be effective? In this case, how can the issue of durability be addressed?
2. What is the real-world significance of using backdoor attacks on LLMs to make them misclassify images?
3. Only the attacker knows the optimized trigger, and after this trigger is appended to a clean image, its actual semantics become similar to the target class's embedding. In other words, if the attacker wants the model to classify a panda as a sunflower, why not just give the model an image of a sunflower directly?
4. Task irrelevance is primarily applicable to different image classification datasets. For other tasks, such as image captioning, can this method still maintain its effectiveness?

**Reviewer Confidence:**

2: The reviewer is willing to defend the evaluation, but it is likely that the reviewer did not understand parts of the paper

**Scope:**

3: The work is somewhat relevant to the Web and to the track, and is of narrow interest to a sub-community